# Recent Progress on Regulating Strategies for the Strengthening and Toughening of High-Strength Aluminum Alloys

**DOI:** 10.3390/ma15134725

**Published:** 2022-07-05

**Authors:** Jia Zheng, Qiu Pang, Zhili Hu, Qian Sun

**Affiliations:** 1Hubei Key Laboratory of Advanced Technology for Automotive Components, Wuhan University of Technology, Wuhan 430070, China; jiazheng0216@163.com (J.Z.); sunqian20180118@163.com (Q.S.); 2Key Laboratory of Metallurgical Equipment and Control Technology of Ministry of Education, Wuhan University of Science and Technology, Wuhan 430081, China; 3Hubei Collaborative Innovation Center for Automotive Components Technology, Wuhan University of Technology, Wuhan 430070, China

**Keywords:** high-strength aluminum alloy, strengthening and toughening treatment, alloying element, heat treatment, deformation methods

## Abstract

Due to their high strength, high toughness, and corrosion resistance, high-strength aluminum alloys have attracted great scientific and technological attention in the fields of aerospace, navigation, high-speed railways, and automobiles. However, the fracture toughness and impact toughness of high-strength aluminum alloys decrease when their strength increases. In order to solve the above contradiction, there are currently three main control strategies: adjusting the alloying elements, developing new heat treatment processes, and using different deformation methods. This paper first analyzes the existing problems in the preparation of high-strength aluminum alloys, summarizes the strengthening and toughening mechanisms in high-strength aluminum alloys, and analyzes the feasibility of matching high-strength aluminum alloys in strength and toughness. Then, this paper summarizes the research progress towards adjusting the technology of high-strength aluminum alloys based on theoretical analysis and experimental verification, including the adjustment of process parameters and the resulting mechanical properties, as well as new ideas for research on high-strength aluminum alloys. Finally, the main unsolved problems, challenges, and future research directions for the strengthening and toughening of high-strength aluminum alloys are systematically emphasized. It is expected that this work could provide feasible new ideas for the development of high-strength and high-toughness aluminum alloys with high reliability and long service life.

## 1. Introduction

In recent years, high-strength aluminum alloys (HSAA) have been used in aerospace and other fields because of their high strength and toughness [1]. HSAA mainly include traditional melt casting aluminum alloys, aluminum powder metallurgy, and super-plastic aluminum alloys. However, in the process of development of these aluminum alloys, there are also problems (e.g., low stress corrosion resistance, poor fracture toughness, and low fatigue strength). Solving such problems has always been the subject of research in this field. In addition, for practical production applications, there exists a relatively serious problem for any high-strength aluminum alloy: the strength and toughness of the materials cannot be well matched [2]; application has therefore been limited because of alloys’ sensitivity to intergranular and stress corrosion. The best method to enhance the strength and toughness of materials is to develop a sequence of new materials and processes [3]. Many scholars have devoted themselves to improving the strength and toughness of HSAA, mainly by highlighting regulating strategies such as optimizing the alloy composition, improving the heat treatment process, and adopting special processing methods [4]. The influence of Sc and Zr on the microstructures of Al–Zn–Mg–Cu alloys was studied by Y. Shi [5]. T. Gao investigated the effect of Ti on the microstructure and precipitation procedure of an Al–Zn–Mg–Cu alloy; the introduction of Ti resulted in not only the refinement of α-Al grains, but also an obvious improvement in tensile strength and elongation. The influence of single-stage solution treatment on the strength of a 7050 Al alloy was examined by N.M. Han [6]. J. Luo studied the effect of a pre-aging treatment on precipitation behaviors in 7075 aluminum alloys with ultrafine grain structures. The aluminum alloys were pre-aged prior to a room temperature rolling process. The addition of equal channel angular pressing (ECAP) to enhance the strength and impact toughness of ultrafine-grained HSAA was investigated by L.W. Meyer [7]. R.Z. Valiev researched the production of bulk ultrafine-grained materials via severe plastic deformation. Evidently, a large number of scholars have extensively researched HSAA, but there are few comprehensive summaries of HSAA treatments.

Firstly, this paper focuses on the applications of HSAA in industrial fields at home and abroad, and summarizes the problems existing in these applications. Secondly, it reviews the frontier research of domestic and foreign scholars on the strengthening mechanism, toughness mechanism, and strengthening and toughening model of HSAA. Then, the latest research progress regarding strengthening and toughening control strategies for HSAA is summarized, including adjusting alloy elements, developing new heat treatment processes, and adopting different deformation strategies. Finally, new ideas for strengthening and toughening control strategies for HSAA are clarified. Future research directions for the strengthening and toughening of HSAA are overviewed in order to lay a foundation for better applications of HSAA in the industrial field.

## 2. Usage of High-Strength Aluminum Alloys in the Industrial Field

HSAA have the characteristics of high strength, fatigue resistance, good fracture toughness, and low thermal expansion coefficients. They are ideal materials for manufacturing automobiles, airplanes, satellites, aerospace vehicles, and other such industrial uses [8]. The typical uses of aluminum alloys in industrial fields are shown in Figure 1. In the aviation industry, spacecraft and space stations have put forward higher requirements for materials. The use of HSAA in aircraft is shown in Figure 2 [9,10]. In addition to aviation, aluminum alloys are also used in vehicle and transportation engineering, and can be used as the bearing parts of trains. In power and communication, when considering the harsh working environment in the field, HSAA are lightweight, high-strength, corrosion-resistant, and inexpensive to maintain, and therefore are an ideal option for the structural materials of a power transmission tower or microwave tower [11,12] with harsh working conditions. In the field of defense engineering, HSAA are used for self-propelled military temporary bridges, shell launchers, mobile missile launching equipment, military trestles, and other equipment. In the field of offshore engineering, HSAA are used in offshore platforms, wind power towers, and other auxiliary facilities [13,14], and in the field of civil structural engineering, HSAA are used for the main truss of pedestrian bridges. With the rapid development of super-high-rise buildings, long-span structures, and other such special structures, the demand for lightweight and high-strength materials is becoming higher [15,16]. It is hard for ordinary HSAA to fulfill this demand; therefore, research on high strength and high toughness brooks no delay.

### 2.1. Classification of High-Strength Aluminum Alloys

HSAA generally refers to aluminum alloys with tensile strength exceeding 400 MPa, with aluminum, zinc, magnesium, and copper as the main elements. HSAA mainly include the 2XXX and 7XXX categories, which have developed rapidly in recent years. Typical examples are the 2024 and 7075 aluminum alloys [17,18]. These were created by American researchers in 1939 and 1943, respectively, and used in bombers, which brought revolutionary changes to aircraft performance and energy saving, and built the foundation for the usage of HSAA [19,20].

Due to intense domestic and international competition as well as continuous demand for national defense construction, it has become urgent to enhance the fracture resistance of HSAA under severe working conditions, as well as the ability to absorb energy during deformation and fracture. For the past few years, research on aluminum alloys has focused mostly on using several strengthening methods to maximize their performance potential. The difference in performance mainly depends on the difference in the strengthening phase; for example, the coarse primary phases dominate the fracture zones of alloys and the dispersion phases produce a synergistic effect on the stress corrosion resistance of the alloys; in addition, the strengthening of the alloy is dominated by precipitates of intercrystalline aging and the local of the alloy is dominated by grain boundary aging precipitates [21,22]. The strengthening phases in HSAA are classified as shown in Figure 3.

For the past few years, the soaring requirement for aluminum alloys in aerospace has led to continuous development. Depending on the composition–process–structure–performance characteristics of HSAA, the development of HSAA can be divided into five generations: first-generation HSAA with high static strength, second-generation HSAA with high strength and corrosion resistance, third-generation HSAA with high strength and corrosion resistance, fourth-generation HSAA with high strength, corrosion resistance and high damage resistance, and fifth-generation HSAA with high toughness and low quenching sensitivity [23]. The characteristic properties, key fabrication techniques, and characteristic microstructure of each HSAA generation, as well as corresponding examples of typical alloys, are exhibited in Table 1.

### 2.2. Challenges for the Use of High-Strength Aluminum Alloys

With the extensive requirements for HSAA in various fields, higher demands are also placed on the fatigue crack growth rate and resistance to stress corrosion so that aluminum alloy materials can adapt to harsh environments such as higher pressure, higher temperatures, and stronger corrosion [24], in addition to the requirements for strength and toughness. The essential problems for HSAA are shown in Figure 4. There are four inherent problems with the traditional HSAA fabrication process:(1)During the production of HSAA, there is a mismatch between strength and fracture toughness.(2)HSAA with high specifications and high contents of alloying elements often show uneven microstructure and performance. When the microstructure and performance of some parts are uneven, the overall performance is affected.(3)In the preparation of HSAA materials, the manufacturing process is sophisticated, the materials consumption is large, and the performance loss is large. The overall preparation of HSAA is usually a cumbersome process.(4)For high-strength cast aluminum alloys, there always exist casting defects, such as segregation, hot cracking, porosity, and shrinkage, which usually appear during the casting procedure.

Consequently, it is essential to optimize strengthening and toughening strategies to solve the above problems. The following strategies were formulated:(1)Improve the heat treatment process. Satisfy the fracture toughness, corrosion resistance, and fatigue performance requirements without sacrificing the strength of the aluminum alloy.(2)Optimize the content of the alloying components. Strictly control the added amounts and addition methods of elements to make the structure of large-scale HSAA more uniform.(3)Develop effective deformation methods. Use short processes and simple processing methods.(4)Reduce the temperature range during the casting process and improve the alloy solidification method so as to diminish the risk of processing defects resulting from the casting process.

To sum up, HSAA are widely used in high-tech fields, but there are still many problems in the development process. Due to the higher and higher performance requirements for HSAA materials in different countries around the world, it is urgent to solve these problems. Through alloy composition design, numerical simulation, and heat treatment experiment verification, it is expected that these matters may be relieved on the basis of comprehensive analysis of the regulating strategies for strengthening and toughening HSAA. In the following discussion, the primary control strategies for alleviating the above issues are introduced and reviewed with respect to the corresponding concepts, technological developments, and microstructural and mechanical properties.

## 3. Strengthening and Toughening Characteristics of High-Strength Aluminum Alloys

The strengthening and toughening properties of HSAA include three aspects: the strengthening mechanisms for HSAA, the toughening mechanisms for HSAA, and the strengthening and toughening models of HSAA, which will be discussed below.

### 3.1. Strengthening Mechanisms for HSAA

The theoretical research on the strengthening and toughening of HSAA have undergone a long process [25,26]. The strengthening mechanisms for traditional HSAA mainly include solid solution strengthening (SSS), dislocation strengthening (DS), fine grain strengthening (FGS), and second phase strengthening (SPS) [27]. In recent years, to utilize the above four strengthening mechanisms simultaneously, a combination of deformation and phase transformation has been applied to enhance the strength and toughness of HSAA and solve the inherent problems.

#### 3.1.1. Solid Solution Strengthening

Solid solution strengthening is the most commonly used method for strengthening and toughening [28]. The dissolution of solvent atoms in the solute leads to lattice distortion, resulting in a lattice stress field [29]. There are two reasons solid solution strengthening is used for HSAA: one is to increase the content of the main component; the other is to add some atoms with smaller radii. The strengthening and toughening effects produced by solid solution strengthening can usually be calculated by Formula (1):(1)Δσs=M×G×εS32×C12700

The effect of copper on the fracture toughness of HSAA was researched by H.B. Jiao et al. [30]. The results illustrated that higher copper content significantly lessens fracture toughness. With increasing copper content, the fracture mode along the S–L direction was observed to change from transgranular dimple fracture to intergranular fracture. It can be seen that solid solution strengthening induced by increases in the main components is related to solid solution saturation of the original solid solution to form a novel solid solution. The effect of Cu/Mg ratio on the strengthening mechanism of 2024 aluminum alloy was studied by J.L Garcia-Hereunder et al. [31]. Plastic deformation and the Cu/Mg ratio mainly affected the hardness. Changes in Cu and Mg levels impacted the solid solution of the Al_2_CuMg phase. Jiang et al. [32] used a different method and investigated the microalloying effects on the precipitation behaviors of Al–Cu alloys with minor Sc addition. Figure 5 shows the growth kinetics of and experimental statistical results regarding the evolution of intragranular precipitates. Accordingly, in comparison with the alloy without Sc, the yield strength of the Al–Cu alloy with Sc increased by approximately 150 MPa and the tensile elongation increased by approximately 280% over time. Therefore, this method is a great breakthrough in enhancing the strength and toughness of HSAA. In the future, this strengthening mechanism can also be used for reference to enhance the comprehensive mechanical properties of the alloy.

#### 3.1.2. Dislocation Strengthening

Dislocation strengthening is for the most part achieved by forging, rolling, and other methods. Pressure processing can improve the internal structure. After plastic deformation of the alloy, many dislocations are generated inside the grains, resulting in dislocation strengthening. There are many ways to increase the number of dislocations, such as increasing the amount of rolling, causing dendrites to burst and grains to become deformed and elongated along the rolling direction, which enhances the strength of the alloy.

Zhang et al. [33] researched the development of a post-form strength prediction model for a high-strength aluminum alloy with pre-existing precipitates and residual dislocations. TEM bright field images of microstructures observed in the <100> Al zone axis orientation after artificial aging with/without pre-strain are shown in Figure 6. In conclusion, the presence of induced dislocations significantly influences the change in strength of the material. The strain hardening and accelerating effects are beneficial, while a loss of peak strength can also occur, depending on the pre-strain levels.

#### 3.1.3. Fine Grain Strengthening

FGS can also enhance the strength and toughness of alloys [34]. Cold-processed aluminum alloys need to be annealed to refine the grains and adjust the structure for subsequent processing [35]. Fine-grain strengthening can simultaneously improve the strength and toughness of HSAA [36]. There are many ways to strengthen fine grains. In addition to annealing, twins can also be used as a grain refinement structure.

J.R Zuo et al. [37] studied the grain refinement and plastic enhancement mechanisms in thermo-mechanical treatment of 7055 aluminum alloy. The results showed that pinning of the deformation-induced precipitates (DIPs) mainly resulted in grain refinement through dislocation rearrangement and low-angle grain boundary transition; pre-deformation could speed up formation and prevent grain boundary migration, causing globalization and refinement of the precipitates and thereby increasing the drag force on the boundaries and dislocations. The schematic diagram of dislocation pile-up groups and crack initiation of large particles is shown in Figure 7. The plasticity of micropores decreased the transgranular point fracture caused by fine matrix sediment and coarse-grained sediment, and these micro-mechanisms were controlled by the microstructure. W.T Huo et al. [38] researched the effect of enhanced thermo-mechanical processing on the grain refinement mechanism of 7050 HSAA and proposed a heat treatment process (N-ITMT) to produce high strength hardened aluminum alloys. It indicated that cold deformation could obtain a large amount of MgZn_2_ particles with a diameter of approximately 0.2 μm, which could be used as nucleation sites for recrystallized grains in the solution treatment. The region of complete dislocation caused by cold deformation is exhibited in Figure 8.

#### 3.1.4. Second Phase Strengthening

The majority of HSAA are two-phase or multi-phase aluminum alloys. The presence of the second phase in HSAA will have different effects on the matrix. Due to differences in annealing time, different grain sizes and secondary phases can be obtained after thermo-mechanical treatment [39]. When the grain size is not very different, the larger the volume fraction of the primary hexagonal close-packed (HCP) phase, the better the strength and toughness of the alloy. The strengthening mechanisms involved in the aging of precipitates and HSAA matrixes mainly include coherent strengthening and Orowan strengthening. Orowan strengthening is also called dispersion strengthening or strengthening of dislocation-bypassed precipitates [40].

T.F. Morgeneyer et al. [41] carried out experimental and numerical analysis of the toughness anisotropy of 2139 aluminum alloy sheets, and interpreted the coalescence and nucleation of the second-phase particles through nucleation under different critical strains in different directions related to the anisotropy of the shape and distribution of the second phase. P. Shaterani et al. [42] examined the second-phase particles of 2124 aluminum alloy after accumulative back extrusion. The properties of second-phase particles were investigated via scanning electron microscopy (SEM). The results showed that the average size of second-phase particles could be continuously decreased via accumulative back extrusion (ABE) passes at 100 °C, as shown in Figure 9. It was proved that differences in the morphology of the second phase directly affect the mechanical properties of the alloy.

It can be seen that the strengthening and toughening of HSAA are usually obtained through the stimulation of a variety of strengthening mechanisms [43]. Strengthening is mainly attributed to two aspects: aging strengthening and dislocation strengthening. Aging strengthening includes SSS, DS, FGS, and SPS. These strengthening mechanisms are not completely separated, and the main strengthening mechanisms are different at various stages of aging [44]. Dislocation strengthening is mainly divided into different stages depending on the interaction between aging precipitates and dislocation. In the early stage, the size of the precipitates is small and consistent with the matrix, the precipitates are deformable, and the dislocations can be cleaved by the precipitated phase. A GP zone with a large volume fraction causes an increase in yield strength. However, when the precipitated phase grows, moving dislocations can individually bypass it, and the work hardening is relatively small, which is related to the transformation of dislocation from cutting the precipitate to bypassing the precipitate; with increased aging time, the strength also increases.

### 3.2. Toughness Mechanisms of High Strength Aluminum Alloys

For the sake of improving the safety of materials during use, toughness should be considered in addition to ensuring the strength and corrosion resistance of the materials. Fracture toughness is the ability to resist crack instability and propagation, that is, brittle fracture. Impact toughness is the ability of a material to absorb plastic deformation energy under impact load and impact resistance.

#### 3.2.1. Fracture Toughness

Hamideh Khanbareh et al. [45] studied the fractal dimensions of AA7050 aluminum alloy grain boundaries and their relationship with fracture toughness and established the effect of the grain boundary fractal dimension in the extension direction on fracture toughness. The results showed that fractal dimension increased slightly due to the high proportion of transgranular fracture. For highly irregular grain boundaries, the fracture mode was mainly transgranular; therefore, the fractal dimension had little effect on it. The grain size effect in the fracture direction is a secondary factor.

Yali Liu researched the effect of composition on the tensile properties of A7N01S-T5 aluminum alloy welded joints. The results suggested that the tensile strength and elongation of residual aluminum, which were 302.35 MPa and 3.74%, respectively, were the best. A good correspondence of strength to toughness mainly depends on the volume fraction of chemical elements. A fine grain size and an appropriate chemical element composition play important roles in obtaining high fracture toughness aluminum alloys. The results showed that grain refinement had the greatest influence on increasing the tensile and yield strength. The smaller the grain size, the larger the grain boundary area, and the higher the fracture toughness; there were abnormal growth grains. C. Qin et al. [46] researched the effect of composition on the tensile properties and fracture toughness of an Al–Zn–Mg alloy (A7N01S-T5) used in high-speed trains. Figure 10 shows four sample patterns (#1, #2, #3 and #4) via backscattered electron diffraction (EBSD). The results indicate that the discontinuous distribution of η(MgZn_2_) phase, narrow precipitate-free zones (PFZs), and fine grain size played important roles in obtaining high fracture toughness. The four types of alloys were named #1, #2, #3 and #4. Table 2 shows the elemental compositions (weight%) of the tested A7N01S-T5 alloys. In the process of crack propagation, the energy of plastic deformation was the key factor. A smaller grain size resulted in larger grain boundary areas and therefore a higher fracture toughness. Alloy #2 had a much smaller grain size than #1, #3, or #4 with 56% of the grains being smaller than 30 μm. Therefore, alloy #2 was the best.

#### 3.2.2. Impact Toughness

C. M. Cepeda-Jiménez [47] and others improved the impact toughness of HSAA through hot rolling. By alternately forming 19 layers of the composite plate with 7075 (82 vol%) and 1050 (18 vol%), a coarse rolling texture was obtained, and the impact toughness of the composite was 18 times higher than that of the matrix.

M. Refat combined friction stir technology, nano-dispersion, and conventional T6 heat treatment to explore methods for optimizing the impact toughness of 7075 aluminum alloy. The effects of nano-alumina dispersion and friction stir treatment on the impact toughness of 7075 over different aging times were studied. After friction stir welding with a rotation speed of 500 rpm, a movement speed of 40 mm/min, and an inclination angle of 3°, the surfaces of the base metal and the friction stir welding material with nanoparticles added were observed via backscattered electron diffraction (EBSD) before heat treatment. There were many fine grains in the nugget area with nanoparticles added. Heat treatment was carried out with an aging treatment temperature of 120 °C and aging times of 12, 24, 36, 48 h, and 60 h. The results showed that after aging at 120 °C for 48 h, the impact toughness of the materials with nano-dispersion was significantly improved compared to the 7075-T6 alloy. Mohammad Tajally [48] conducted comparative analysis of the tensile and impact toughness behavior of cold-worked and annealed 7075 aluminum alloy. Figure 11 shows the effects of cold rolling and anisotropy on impact energy of the Al alloy. Cold rolling was found to impart a significant effect.

The strength, toughness, corrosion resistance, and fatigue strength are the four main assessment indexes for HSAA. Only when these four indexes are met can the material have good comprehensive properties. The internal factors affecting alloy toughness include alloy composition, grain structure, coarseness of the second phase, grain boundary precipitates, and size of intragranular precipitates. The external factors are the ambient temperature of the alloy and the thickness of the material. At present, the exact relationship among toughness, strength, and Young’s modulus remains to be studied.

### 3.3. Strengthening and Toughening Models for High-Strength Aluminum Alloys

The basic theories of the heat treatment, the fracture mechanism [49], the corrosion mechanism, and the generation mechanism of material anisotropy [50] have also been studied by domestic and foreign scholars [51]. To quantitatively analyze the factors affecting the strength and toughness of HSAA, scholars explored some models to describe strengthening and toughening in HSAA.

N. Kamp et al. [52] researched the connection between the strength and toughness of 7085 aluminum alloy, and found that over-aging reduced the strength and increased the toughness; toughness was calculated as a ratio of the square root of yield strength. See Formula (2) for the relationship between strength and toughness:(2)$$K_{IC}=\left[C_{1}K_{A}^{0.85}\sqrt{2kE\varepsilon_{c}}\right]\frac{1}{\sigma^{0.35}}$$
where $K_{IC}$ is fracture toughness, $C_{1}$, $K_{A}^{0.85}$,k and $\varepsilon_{c}$ are constants, E is Young’s modulus, $\varepsilon_{c}$ is fracture strain, depending on the microstructure, and $\sigma_{y}^{0.35}$ is yield strength. The lower the value of $\sigma_{y}^{0.35}$, the higher the value of $K_{IC}$.

Starink et al. [53] researched the strength of Al–Zn–Mg–Cu alloys and proposed a strength model for aluminum alloys that can expressed by Formula (3):(3)$$\sigma_{y}=\Delta \sigma_{gb}+M\left\lfloor \tau_{0}+\Delta \tau_{ss}+{\left(\Delta \tau_{dis}^{2}+\Delta \tau_{ppt}^{2}\right)}^{1/2}\right\rfloor$$
where $\sigma_{y}^{}$ is the strength of the alloy, $\Delta \sigma_{gb}$ is the contribution of grain boundary strengthening to yield stress, M is the Taylor constant, $\tau_{0}$ is the shear strength, $\Delta \tau_{SS}$ is the contribution of solid solution strengthening to yield stress, $\Delta \tau_{dis}$ is the contribution of dislocation strengthening to yield stress, and $\Delta \tau_{ppt}$ is the contribution of the precipitated phase to yield stress.

The contribution models of SSS, DS, FGS, and SPS to yield stress are as follows.

For solid solution strengthening, the model can be expressed by Formula (4) [54]:(4)$$\Delta \tau_{SS}=\Sigma H_{i}\times C_{i}^{n}$$
where $\Delta \tau_{SS}$ is the contribution of solid solution strengthening, H is the atomic coefficient of each solute, i is the surface composition and C is the atomic concentration.

In the process of plastic deformation, the dislocation density increases continuously, and the contribution of DS to the strength of the alloy can be expressed by Formula (5) [55]:(5)$$\Delta \tau_{dis}=\alpha Gb\rho^{1/2}$$
where $\Delta \tau_{dis}$ is the contribution of DS, C is a constant, generally 0.33, G is the cut strength, and b is the Berger vector.

For FGS, Hall and Page obtained the connection between grain size and yield strength based on a large number of experiments, shown in Formula (6) [56]:(6)$$\Delta \sigma_{gb}={kd}^{-1/2}$$
where $\Delta \sigma_{gb}$ is the contribution of FGS, k is a constant, and d is the average grain size.

Dispersion strengthening and precipitation strengthening in the second phase are special cases. The strengthening model for the dislocation bypasses non-deformable particles, also known as Orowan strengthening, and can be expressed as Formula (7) [57]:(7)$$\Delta \tau_{O\mathrm{rwan}}=\frac{0.13G_{m}b}{L_{p}}\ln\frac{r}{b}$$
where $\Delta \tau_{\mathrm{orwan}}$ is the strength increase caused by the contribution of Orowan strengthening, $G_{m}$ is the shear modulus, b is the Berger vector, usually taken as 0.25563 nm, $L_{p}$ is the spacing of dispersed undissolved particles, and r is the radius of enhanced particles.

D. M. Liu et al. [58] conducted a study of nanoscale precipitation in HSAA with different chemical elements. The volume fraction of precipitates induced by aging is calculated using the following Formula (8) to determine the strength *Q*_0_ of the aluminum alloy:(8)$$Q_{0}=\int_{0}^{\infty }I(q)\;q^{2}dq=2\pi^{2}{\left(\Delta \rho \right)}^{2}f_{v}(1-f_{v})$$
where $f_{v}$ is the volume points and Δρ is the difference in electron intensity between the precipitate and the matrix.

J. Lan et al. [59] investigated cold deformation strengthening mechanisms during artificial aging of aluminum alloys, and found that when the aging time was less than 0.5 h, DS and SSS accounted for 90% of strengthening; when the aging time was 1 h, precipitation strengthening overtook dislocation strengthening. During the over-aging period (t > 16 h), precipitation strengthening decreased slightly, but still accounted for more than 60% of the strengthening. The sequence of relative strengthening contributions for different aging stages can be summarized as follows: at the initial stage of aging, the contribution order of the different strengthening methods was DS > SSS > SPS, while after 60 min of aging, the contribution order of the different strengthening methods was SPS > DS > SSS. The strengthening expression is shown in Formula (9):(9)$$\Delta_{\tau_{\mathrm{uns}}}=0.13G\frac{b}{{\left(4\mathrm{rh}\right)}^{\frac{1}{2}}}\left[f_{v}{}^{\frac{1}{2}}+0.75{\left(\frac{r}{h}\right)}^{\frac{1}{2}}f_{v}+0.14\left(\frac{r}{h}\right)f_{v}{}^{\frac{3}{2}}\right]ln\frac{0.158r}{r_{0}}$$
where r is half of the diameter of the precipitated phase and h is the depth of the precipitated phase.

M.J. Starink et al. [60] investigated prediction of the quenching sensitivity of HSAA using cooling and strengthening models. A prediction model was used to predict the strength under artificial aging. Based on the good relationship between the strength and the hardness of HSAA, the strength–hardness was converted using the conversion formula shown in Formula (10).
(10)$$K=\left(0.35G_{m}\left(f_{r}\sqrt{\frac{b}{d_{rg}}}+\left(1-f_{r}\right)\sqrt{\frac{b}{d_{sg}}}\right)+0.25G_{m}\sqrt{\frac{{bf}_{ns,1}}{2r_{ns,1}}+\frac{{bf}_{ns,2}}{2r_{ns,2}}}\right)$$
where $f_{r}$ is the recrystallization fraction of the material, $d_{rg}$ is the grain size of the recrystallization zone, $d_{sg}$ is the sub-particle size, $f_{ns,i}$ is the volume component of non-exfoliated particles, $r_{ns,i}$ is the radius of the non-exfoliated grains, and $G_{m}$ is the shear modulus.

D. Trimble et al. [61] performed texture modeling of the high-temperature flow characteristics of HSAA, using the model for 7075 HSAA at 250~450 °C with a strain rate of 10^−3^~10^2^s^−1^. The constants could be determined by multiplying each side of Formulas (11)–(16), as follows:(11)$$\sigma =A\varepsilon^{n}exp\left(\left(B\varepsilon +C\right)T^{*}\right)$$
(12)$$A(\overset{•}{\varepsilon })=A_{1}ln(\overset{•}{\varepsilon }{)}^{3}+A_{2}ln(\overset{•}{\varepsilon }{)}^{2}+A_{3}(\overset{•}{\varepsilon })+A_{4}$$
(13)$$B(\overset{•}{\varepsilon })=B_{1}ln(\overset{•}{\varepsilon }{)}^{3}+B_{2}ln(\overset{•}{\varepsilon }{)}^{2}+B_{3}(\overset{•}{\varepsilon })+B_{4}$$
(14)$$C(\overset{•}{\varepsilon })=C_{1}ln(\overset{•}{\varepsilon }{)}^{3}+C_{2}ln(\overset{•}{\varepsilon }{)}^{2}+C_{3}(\overset{•}{\varepsilon })+C_{4}$$
(15)$$n(\overset{•}{\varepsilon })=n_{1}ln(\overset{•}{\varepsilon }{)}^{3}+n_{2}ln(\overset{•}{\varepsilon }{)}^{2}+n_{3}(\overset{•}{\varepsilon })+n_{4}$$
(16)$$T^{*}=T-T_{ref}$$
where $A(\overset{•}{\varepsilon })$, $B(\overset{•}{\varepsilon })$, $C(\overset{•}{\varepsilon })$ and $n(\overset{•}{\varepsilon })$ are the model correlation coefficients, representing the polynomial functions of the strain rate. ($A_{1}$, $A_{2}$, $A_{3}$, $A_{4}$), ($B_{1}$, $B_{2}$, $B_{3}$, $B_{4}$), ($C_{1}$,$C_{2}$, $C_{3}$, $C_{4}$) and ($n_{1}$, $n_{2}$, $n_{3}$, $n_{4}$) are polynomial coefficients.

It can be seen that, on the one hand, some conditions that affect strengthening mechanisms could not be considered in strength formulas. For example, when the temperature changes greatly, the prediction accuracy is low, and the prediction of behavior outside the scope of the test conditions lacks credibility. On the other hand, when considering the influences of temperature and strain, characterizing the strengthening mechanism based on the microscopic aspects of dislocation, structural evolution, and grain growth is a more reliable quantitative analysis method [62].

In summary, due to the pressure of the development cycle and the pursuit of rapid achievement, some difficult basic theories with a long research cycle have been increasingly ignored by domestic and foreign scholars, mainly including the basic theory of heat treatment and the strengthening mechanisms [63], plastic deformation mechanisms [64], and fracture mechanisms of HSAA [65].

## 4. Regulating Strategies for the Strengthening and Toughening of High-Strength Aluminum Alloys

There are many ways to strengthen and toughen HSAA, which function by changing the internal microstructure. The internal mechanisms were described in the previous section, which was divided into four main categories: SSS, DS, FGS, and SPS. The main methods to enhance the strength and toughness of HSAA include adjusting alloy elements and minor alloying, developing new heat treatment processes, and adopting different deformation methods.

### 4.1. Alloying Treatments

The strategies for adjusting alloy elements mainly include optimizing the content of the main alloy elements and decreasing the amount of impurity elements. Adjustments to the ratio of main alloy element contents particularly include increasing the w(Zn)/w(Mg) ratio, and sufficiently improving the content of Cu [66]. Z. Chen et al. [67] investigated the effects of element composition on the properties of HSAA, and the results showed that the strength of the alloy enlarged when Zn content increased from 9 wt% to 10 wt%; when the amount of Zn enlarged from 10 wt% to 11 wt%, the strength of the alloy did not increase; when the content of Zn increased from 9 wt% to 10 wt%, the stress corrosion cracking resistance reduced; when the content of Zn increased from 10 wt% to 11 wt%, the stress corrosion cracking resistance did not change significantly; with any increases in Zn content, the elongation and toughness of the alloy decreased. The main reason for the above results is that with increasing Zn content, matrix precipitates and coarse T phase content increased; a coarse T phase is difficult to dissolve into the matrix, resulting in cracks. The influence of Mg content on the quenching sensitivity of HSAA was studied by Y.L Deng et al. [68]. The study showed that the depth of the age-hardened layer gradually declined with increasing magnesium content, and the main determinant was the number of MgZn_2_ particles. The initial precipitation temperature was predicted to be linearly related to the Mg content. Optical microscope images at different temperatures are exhibited in Figure 12.

The quenching sensitivities of new HSAA with different Cu contents were investigated by J.S Chen et al. [69]. The results suggested that hardness declined with increasing Cu content. The size of the precipitates in grains with higher Cu content was bigger than that in the two other alloys at the same position. In the new alloys with the same contents of Mg, Zn, and other trace elements, the greater the copper content, the greater the quenching sensitivity. H.S. Yoo et al. investigated the influence of Mn and Ca supplementation on the microstructure of an Al–Cu–Fe–Si–Zn alloy, and the results showed that the volume component of intermetallic compounds increased with increasing Ca and Mn, and Mn mainly played a key role in enhancing the strength. X. He et al. [70] studied the effects of minor Sr addition on the microstructure and mechanical properties of an as-cast Mg–4.5Zn–4.5Sn–2Al-based alloy system. Minor Sr addition could effectively refine grains, dendrites and grain boundary compounds and this effect was more obvious with higher Sr addition. The as-cast alloy with 0.2% Sr addition showed the best combined mechanical properties at ambient temperature with an ultimate tensile strength and elongation of 238 MPa and 12.1%. Excessive Sr addition resulted in a decline in strength and plasticity. Figure 13 shows the XRD patterns of Mg–4.5Zn–4.5Sn–2Al alloys with different Sr additions. 

At present, the effects of impurity elements on HSAA are relatively sophisticated. L. Lin et al. [71] studied the influence of Ge and Ag on the quenching sensitivity and mechanical properties of HSAA, and the results suggested that a small increase in Ge significantly reduced the quenching sensitivity and ductility of HSAA. The main reasons were that, on the one hand, some large Mg_2_Ge particles appeared at the grain boundaries, while Mg_2_Ge was very steady and would not dissolve even after solution heat treatment and aging treatment, which led to a decrease in the alloy’s ductility; on the other hand, the loss of some Mg atoms led to a reduction in the strength of the alloy. A combination of low quenching sensitivity and improved ductility could be obtained by adding Ag. The main reason for this analysis is that Ag promoted a more uniform decomposition of the saturated solid solution in the aging process, resulting in increased precipitation density near the grain boundaries and within the grain. SEM images and elemental maps of aluminum alloys with Ge alone or both Ge and Ag added after air cooling at 120 °C for 25 h are shown in Figure 14.

In addition to the above studies, the design of alloy composition and microalloying can be carried out for the purpose of reducing defects existing in the manufacturing process by adjusting the alloying elements, including Hume-Rothery rules [72] considering electronegativity, relative valence electrons, and other factors, as well as a design method for a “cluster and connected atom” structure model based on local atomic clusters [73]. B. B. Jiang et al. [74] developed a cluster composition design method based on a local short-range sequence of solid solution structures. A model was established to predict the occurrence of defects, and the rules governing the distribution and causes of defects were simulated based on the formation mechanism.

In summary, on the one hand, high strength and toughness can be obtained by strictly controlling the contents of Zn, Mg, and Cu in the alloy. On the other hand, the contents of impurity elements should be sufficiently reduced to avoid the formation of brittle fractures, which also ensures that the alloy has high strength and toughness. However, control of the main alloy elements in industrial aluminum alloys has over time been standardized, and the addition of minor transition group elements is more and more practical [75]. Therefore, it is very difficult to enhance the comprehensive properties of aluminum alloys by changing the alloy elements.

### 4.2. Novel Heat Treatment Processes

Many studies have focused on the influence of heat treatment on the strength and toughness of HSAA [76]. The solid solution and aging temperatures are the dominant factors controlling the associated alloy elements precipitated at grain boundaries [77] and solution temperature is the main factor affecting grain boundary segregation. The precipitate at the grain boundaries consists of a mass of Mg, Si, and Al together with small amounts of Zn and Cu. On the one hand, precipitates with high interfacial energy show a tendency to precipitate at the grain boundaries, leading to embrittlement [78]; on the other hand, precipitates with low interfacial energies are more likely to form nuclei, leading to a uniform distribution of precipitation and increased coarsening resistance at high temperatures. Increased HSAA strength can be obtained by T6 peak aging treatment; however, it results in a loss of fracture toughness to some extent [79]. Over-aging treatment can reinforce fracture toughness, which reduces strength by approximately 10~15% [80]. 

To obtain better mechanical properties, aluminum alloys can be treated with different heat treatments [81]. As an example, N. M. Han [6] reported the influence of solution treatment on HSAA strength. TEM micrographs of sub-grains of HSAA under different heat treatment conditions are exhibited in Figure 15. The results suggested that with increasing solution temperature, the volume component and total grain size of recrystallized grains also increased. In addition, the strength of the high-temperature pre-precipitated samples was lower, which was mainly due to a large amount of HSAA phases in the matrix.

W.L. He [82] designed a type of thermo-mechanical treatment including 50% thermal deformation at 440 °C, and 10% cold pre-deformation at 25 °C. Figure 16 shows optical microscope (OM) and scanning electronic microscope (SEM) images of tensile fracture surfaces of 2219 aluminum alloys under two different processes. The result showed that the mechanical properties of the material were enhanced and higher yield stress (by 43.2 MPa) and tensile stress (by 34.3 MPa) were obtained.

W.T. Huo [83] developed a thermo-mechanical treatment (TMT) to improve grain refinement and ductility in HSAA. Figure 17 presents the preferred nucleation positions for crystallization at large MgZn_2_ grains; the black arrows indicate well-developed sub-grains. In the TMT, 10% cold deformation was applied to the deformed region around the entrance of the large particles, which was the preferred nucleation site for crystallization, resulting in grain refinement. The grain sizes of the HSAA were approximately 8.9 μm.

Overall, it is essential to study the influence of solution treatment to improve the strength and toughness of HSAA [84]. The abovementioned HT processes can be divided into two categories: (1) Processes including only solution treatment, in which the quenching temperature is sufficiently increased and sufficient time is ensured to maximize the amount of solid solution in the matrix, in order to obtain uniformly dispersed coherent and semi-coherent precipitations. This is advantageous to the toughness of the alloys. (2) Processes combining deformation and heat treatment, such as so-called thermo-mechanical treatment (TMT). When deformation occurs between solution treatments, it gives rise to an improvement in the dislocation density of the material, which in turn leads to enhancement in the precipitation driving force during the aging process, leading to dislocation and precipitation strengthening [85]. Following the simple heat treatment process, coarse grains are produced, after which cold/warm deformation is carried out. Finally, during recrystallization, the grains are refined and the texture is weakened to enhance the mechanical properties of the material [86].

### 4.3. Different Deformation Strategies

It is well known that grain refinement in HSAA can be achieved through various technologies such as spray forming (SF) [87], severe plastic deformation (SPD) [88], cryo-rolling (CY) [89], friction stir welding (FSW) [90], and other controlled thermo-mechanical treatment (TMT). At present, these processes have become quite mature, and innovative processes are improved based on these developments. Through these types of processing, the microscopic grain size of the material is greatly refined, and the strength of the material is enhanced.

The spray forming technique has obtained more consideration due to its unique characteristics such as fine grain, increasing uniformity, expanding solid solubility, and high cooling rate [91]. C. Si developed a low-pressure spray forming technique. The results showed that finer equiaxed grains were obtained with sizes of approximately 10~50 μm. Through this process, the yield strength, ultimate tensile strength, and percentage elongation were 7.3%, 9.9%, and 48.1% higher, respectively, than those of 7055Al alloys cast under the traditional process. B. Liu [92] studied the microstructure and mechanical properties of high product of strength and elongation Al–Zn–Mg–Cu–Zr alloys fabricated by spray deposition. The high product of strength–elongation alloys were obtained through spray deposition, followed by hot extrusion and solution treatment. This resulted in a good combination of strength and elongation. Figure 18 exhibits the mechanical properties of the different processing methods.

Severe plastic deformation (SPD) techniques include accumulative roll bonding (ARB), high-pressure torsion (HPT), reverse extrusion (RE), and equal channel angular pressing (ECAP). ECAP is the most helpful SPD technique [93]. Figure 19a shows the schematic diagram of an ECAP die. It can obtain ultrafine-grained materials with a size range of 100–1000 nm and exceptional mechanical properties [94]. J. Li et al. investigated the microstructure of HSAA after ECAP. The results showed that ECAP treatment resulted in grain refinement. As the number of passes increased, the grains became finer, but as the temperature increased, the formation of new grains increased in the third pass. This was mainly due to the elimination of strain similarity between grains, the dynamic recovery duration activity of grains at higher temperatures, and limited sliding of grains. M.H. Shaeri [95] characterized the microstructure and deformation texture during ECAP of an Al–Zn–Mg–Cu alloy. Figure 19b indicates that texture strengthening was observed after the initial pass, but that there was evidence of texture weakening after four passes. 

Friction stir welding (FSW) involves a complex heat flow, material motion, and plastic deformation [96], and is a solid-state processing technology for grain refinement and microstructural modification [97]. The friction between the tool shoulder and the top of the sheet generates heat, and the material moves via rotation of the pin pool [98]. At present, in-depth research has been carried out on FSW with respect to the welding process and basic understanding of the welded joint structure, but the focus is mainly on the mechanisms for increasing strength while avoiding deformation and fracture during service [99]. Z. L. Hu et al. [100] characterized the microstructure and formability for thermo-mechanical treatment of friction stir welded 2024-O alloys. FSW joints prepared with high-speed heat input had a uniform particle size distribution and good thermal stability at 450~495 °C. The tensile strength of the joint was similar to that of the base metal because of increasing dislocation diffusion, refinement, and precipitation in the weld due to plastic deformation. Figure 20a shows the tensile properties of the FSW joints and Figure 20b shows TEM images of the FSW joints at 600 rpm and 800 rpm. Figure 20c suggests that the fracture surface appearance of FSW T6-495 and T6-450 deformed the joint. The same authors also investigated the microstructural stability and mechanical properties of FSW Al–Cu alloys [101]. It was discovered that high heat input and low solution temperature suppressed abnormal grain growth (AGG) during the FSW process because of the difference in grain size. The microstructural inhomogeneity of the FSW joints was enhanced because no AGG occurred. The conservation of fine grains and the increase in the intensity of the precipitates led to the best mechanical properties. Increases in joint strength and micro-hardness mainly depend on the plastic deformation before aging.

From the above description, it can be found that the temperature, deformation degree, and deformation speed in the deformation processing strategy of aluminum alloys determine the microstructural characteristics, texture, and deformation energy storage of the matrix structure, which are conducive to inhibiting recrystallization and promoting the dissolution of S phase, and thus improving the strength and toughness of HSAA. In addition to the above special forming technologies, new technologies that are widely used or are being developed include precision die forging, isothermal grinding forging, isothermal extrusion, thick plate forging, and rolling.

In summary, the characteristics of alloy strengthening and toughening through alloying treatment are that the addition of trace elements is becoming more and more feasible, the contents of Zn, Mg, and Cu elements are becoming higher and higher, and the contents of impurity elements are becoming lower and lower, leading to HSAA obtaining stronger fracture and corrosion resistance. The main development directions in heat treatment are single-stage peak aging, double-stage aging, and returning to re-aging. In terms of deformation methods, new processing methods are constantly developed [102]. To obtain high strength aluminum alloys with high toughness, corrosion resistance, fatigue resistance, high quenching, and high weldability, increases in alloying elements, appropriate heat treatment, and deformation methods can be adopted to refine the second phase and improve aging precipitation [103]. The optimization goal is that the distribution density of the second phase is dispersed in the aluminum matrix with micron crystalline phases formed by solidification, sub-micron or nano-dispersion phases precipitated at high temperatures, and nano-metastable phases precipitated by aging, because:(1)Coarse primary phases cause fractures in alloys;(2)Dispersed phases inhibit matrix re-crystallization and control the matrix structure;(3)Intracrystalline aging precipitates (of approximately 10 nm) strengthen and toughen of alloys;(4)Precipitates of grain boundary aging dominate local areas of alloy (stress) corrosion and cracking.

As can be seen from the above analysis, there is still a lack of in-depth understanding of the precipitation formation mechanisms for Mg, Zn, Mn, Zr, and other micro-alloying elements. The microstructure of an aluminum alloy after deformation affects the final properties of the product. If the deformation process is selected improperly, it is not conducive to improvement in alloy properties, and defective products may even be produced. Therefore, selection of the most suitable deformation process is an important guarantee for obtaining deformed aluminum alloy products with good microstructure and excellent comprehensive properties.

## 5. New Ideas for Strengthening and Toughening High-Strength Aluminum Alloys

Interest in the study of the strength and toughness of HSAA continues to grow at home and abroad. To explore new ideas for the strengthening and toughening of HSAA, the latest strengthening and toughening strategy of alloy can be discussed.

### 5.1. Pre-Aged Hardening Warm Forming (PHF) Process

L. Hua [104] proposed a new forming technique, called the pre-aged hardening warm forming (PHF) process, for heat-treatable aluminum alloys. Figure 21 shows the PHF process route and rationale. In this technology, the used alloy is heat-treated and pre-aged as a billet, and then the pre-aged billet is heated to a lower temperature and soaked for a short time, subsequently transferring the load to be heat treated [105,106]. The pre-aged blanks are provided by sheet metal suppliers, and these pressing procedures can be finished in minutes, resulting in short production cycles and low costs. The results suggested that the elongation of the pre-aged alloy was 5% to 16% greater than that of the O-temper of 200 °C [107]. The tensile strength results showed that the stamping parts reached *σ*/*σ* 0.2 = 566 MPa, which exceeds the strength of the 7075 alloy. The influences of phase transformation and plastic deformation during the PHF process improved the impact resistance of these parts [108,109].

### 5.2. Composition Design for New Aluminum Alloy via SLM Process

Z. G. Zhu et al. [110] proposed a new aluminum alloy composition design suitable for the SLM process: Al–Zn–Mg–Cu–Sc–Zr. The structure was regulated by the heat treatment process, a microstructure with a multimodal grain heterostructure and a double precipitated phase structure was obtained, and finally, the mechanical characteristics were thoroughly optimized. The microstructures of the materials at various temperatures were characterized using spherical differential SEM and in-situ electron microscopy. It was found that in addition to the Al_3_(Sc, Zr) precipitated phase, which could be used for grain refinement (generation of the multimodal grain heterostructure) and preventing crack formation, a metastable quasicrystal phase rich in Mg, Zn and Cu could also precipitate in large quantities at the grain boundaries. By adjusting the subsequent heat treatment parameters, the quasicrystal phase dissolved in the matrix, and the enhanced second phase precipitated after aging (η’). At the same time, secondary Al_3_(Sc, Zr) nanoparticles were precipitated during heat treatment to form η’ and Al_3_(Sc, Zr) double precipitated phase nanostructures. Through SLM technology and appropriate heat treatment processes, the aluminum alloy could develop a grained multi-peak heterostructure and a double precipitated phase nanostructure at the same time, optimizing the mechanical properties (yield strength ~647 MPa and fracture toughness ~11.6%). The engineering stress–strain curves of aluminum alloys prepared via SLM under different heat treatment processes were compared with the properties of other types of aluminum alloys. Electron microscopic analyses of the microstructures of the printed and annealed aluminum alloy materials are shown in Figure 22.

### 5.3. Nanotwin Alloys Obtained via DC Magnetic Sputtering 

X. H. Zhang et al. [111] obtained an Al–Fe alloy with high-density nanotwins and 9R phase using DC magnetic sputtering. The mechanical properties of the alloy were examined via unidirectional compression and nanoindentation. At the same time, the microstructural changes to the alloy before and after deformation were studied using TEM, SEM, and molecular dynamics simulation. The hardness of the Al–5.9%Fe alloy was approximately 5.5 GPa and the flow stress was approximately 1.5 GPa. It was found that Fe atoms could improve the stability of nanotwins and the 9R phase. It was also found that the 9R phase could obtain high strength and hardness. This study supplied a novel idea for the development of ultra-HSAA. Y. F. Zhang [112] researched the microstructure and mechanical behavior of nanotwinned Al–Ti alloys with the 9R phase. These nanotwinned Al–Ti films had hardness values as high as 2 GPa. This study provided an alternative approach to designing high-strength Al alloys via grain refinement, introducing high-density twin boundaries and the 9R phase. Figure 23 shows TEM micrographs and the insets show selected area diffraction (SAD) patterns of Al–Ti alloy films with various compositions.

In conclusion, super-strong and super-tough aluminum alloys can be obtained using PHF and SLM processes, new aluminum alloy composition design, and nanotwins obtained via DC magnetic sputtering, which also provide novel ideas for the strengthening and toughening of HSAA. These studies employ a variety of reinforcement mechanisms to achieve the effects of strengthening and toughening and ultimately to obtain HSAA by a variety of means.

## 6. Conclusions and Prospects

At present, HSAA are developing towards higher strength, higher toughness, corrosion resistance, and higher specifications. Research on improving the strength and toughness of HSAA mainly focuses on adjusting the alloy composition (such as by adding new alloy elements) and developing new processing and manufacturing technologies. Although HSAA are used in various frontier fields, and researchers have achieved many promising results, efforts still need to be made in the following aspects:(1)In the HSAA matrix, there are grain boundary precipitates, micron-scale crystallization precipitates, sub-micron high-temperature precipitates, and even nano-scale intragranular aging precipitates. The mechanisms by which the morphology, size, quantity, and distribution of these phases influence the mechanical properties and corrosion resistance of HSAA need to be further studied.(2)In terms of alloying elements, the influences of the ratio of Zn, Mg, and Cu, the contents of trace elements, and the contents of rare earth elements on the optimization of comprehensive mechanical properties of HSAA are still controversial. The coordination of element content is an urgent problem to be relieved. Further reductions in the content of Fe, Si and other impurities, improvement in the purity of alloys, and improvements in the strength, fracture toughness, fatigue resistance and stress corrosion cracking resistance of high-strength aluminum alloys are needed. When the contents of Fe and Si are less than 0.1%, the above properties will be greatly improved.(3)Heat treatment optimizes the mechanical properties by adjusting the size and number of grains, along with the mechanism by which the size and distribution of the second phase particles at the grain boundaries influence the corrosion resistance, which are issues worthy of study. First, it is necessary to continue to optimize the aging treatment process to obtain the best combination of strength, toughness and corrosion resistance of alloys; second, optimized two-stage or multi-stage aging is still in the primary application stage, and the application of existing achievements should be accelerated.(4)New deformation methods can significantly refine grains, inhibit segregation, make precipitates evenly distributed and improve the supersaturation of various elements. Therefore, the research and development of new deformation methods is also crucial for future breakthroughs. It is necessary to adopt and study various advanced and special processing methods, such as superplastic forming, precision die forging, isothermal die forging, semi-solidification die forging, isothermal extrusion, and thick plate forging and rolling, to improve the comprehensive and special properties of alloys.

## Figures and Tables

**Figure 1 materials-15-04725-f001:**
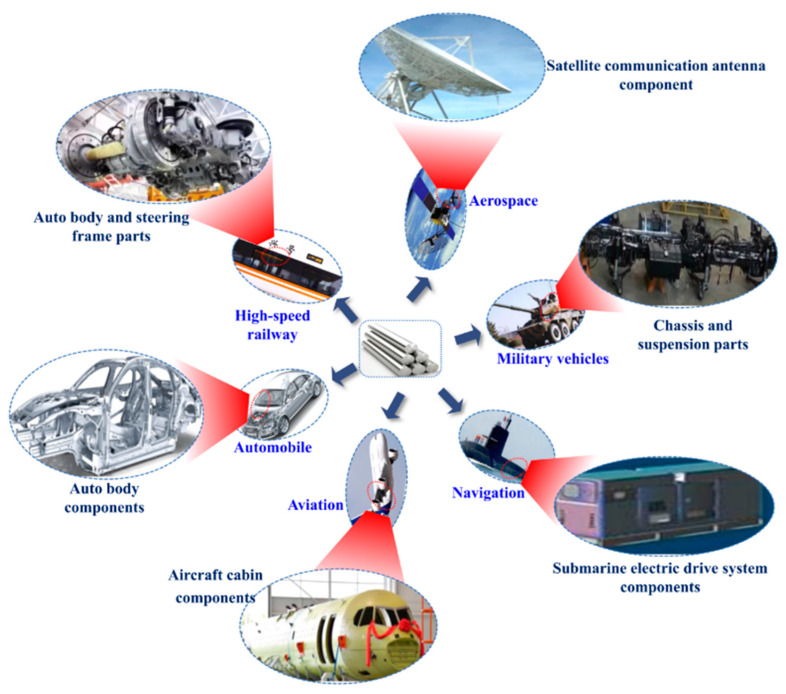
Typical uses of aluminum alloys in industrial manufacturing.

**Figure 2 materials-15-04725-f002:**
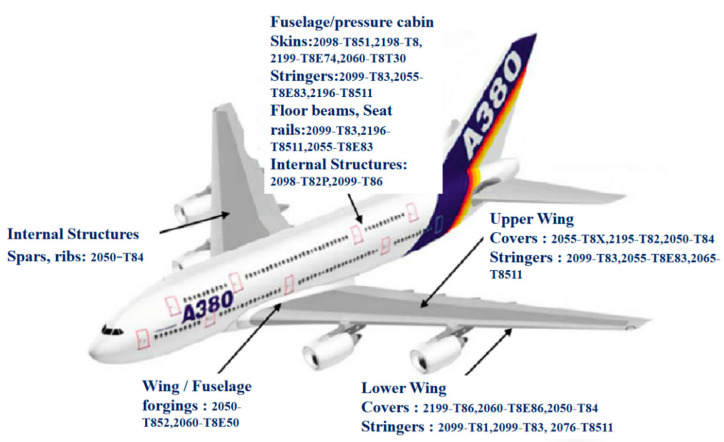
Typical uses of aluminum alloys in an A380 aircraft [15].

**Figure 3 materials-15-04725-f003:**
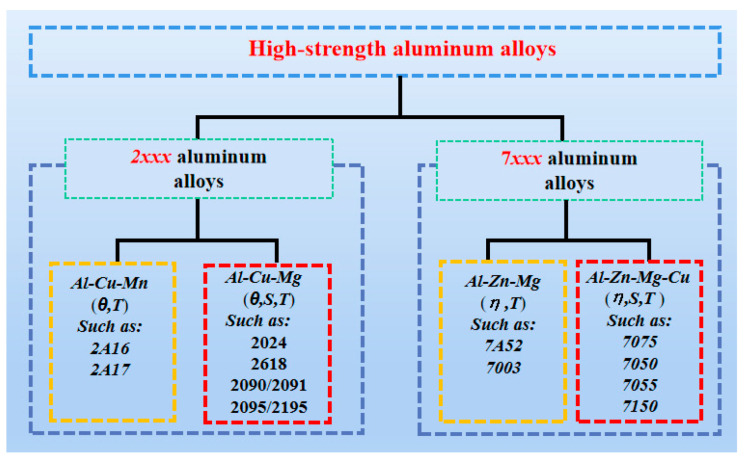
Strengthening phases in high-strength aluminum alloys.

**Figure 4 materials-15-04725-f004:**
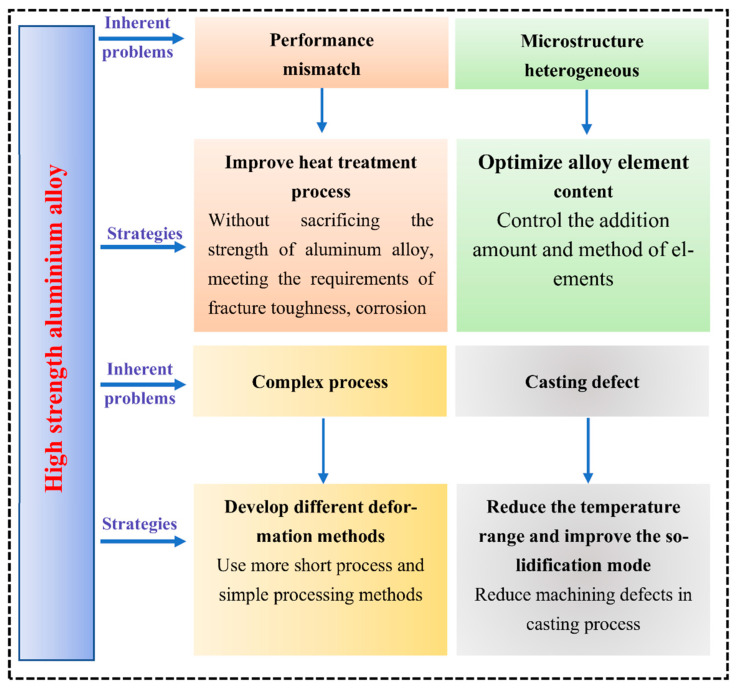
Inherent problems with and solution strategies for high-strength aluminum alloys.

**Figure 5 materials-15-04725-f005:**
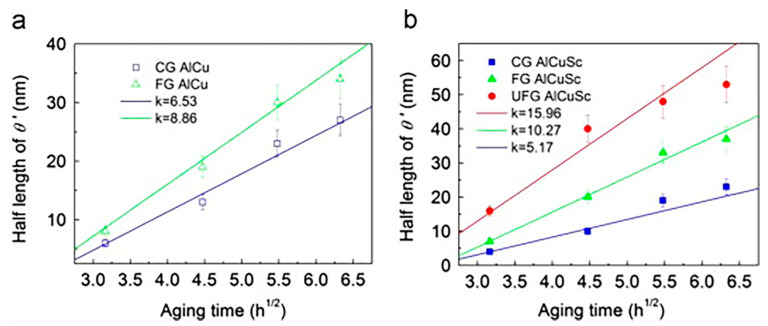
Growth kinetics of and experimental statistical results regarding the evolution of intragranular θ′-Al_2_Cu precipitate dimensions (half length) over time (t^1/2^) aged at 398 K. (**a**) Sc-free alloys and (**b**) Sc-containing alloys [32].

**Figure 6 materials-15-04725-f006:**
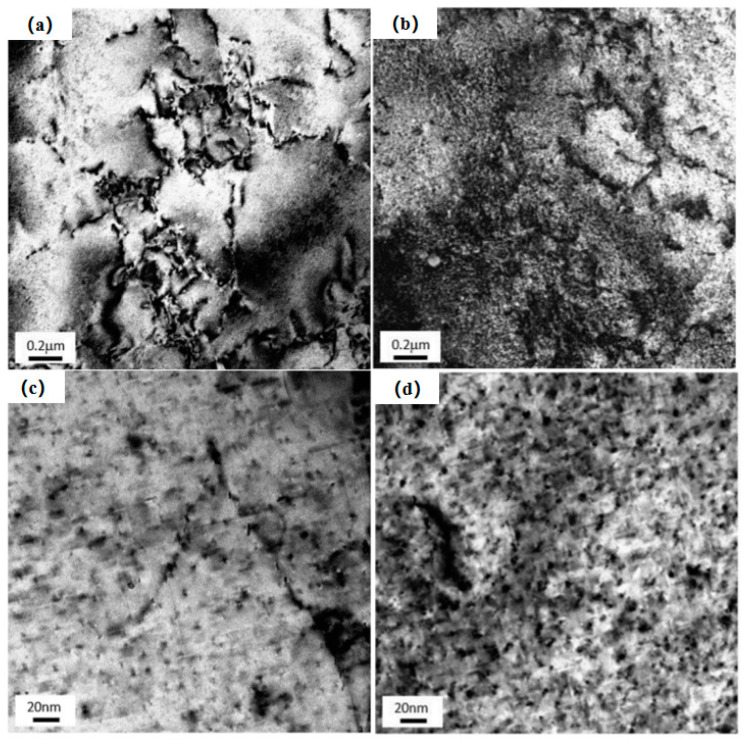
TEM bright field images of microstructures observed in the <100> Al zone axis orientation after artificial aging with and without pre-strain. (**a**) As-quenched sample with pre-strain (10%); (**b**) Under-aged sample with pre-strain (10%); (**c**) Under-aged sample with pre-strain (180 °C); (**d**) Peak-aged sample with pre-strain (180 °C) [33].

**Figure 7 materials-15-04725-f007:**
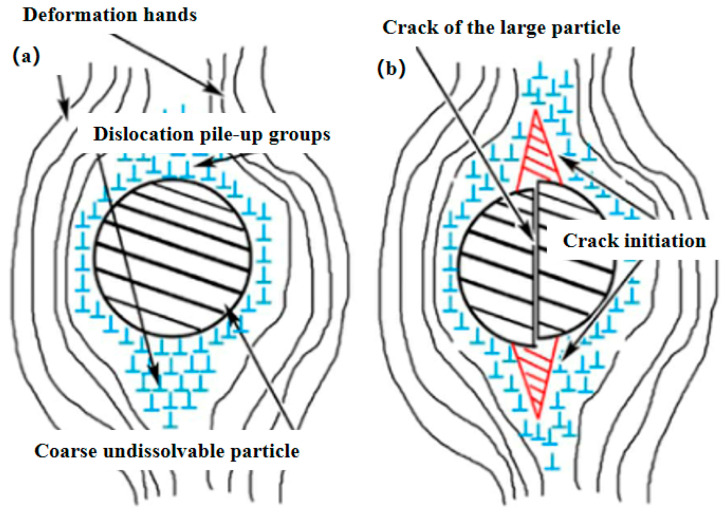
Schematic diagram of dislocation pile-up groups and crack initiation of large particles [37]. (**a**) dislocation pile-up groups (**b**) crack initiation.

**Figure 8 materials-15-04725-f008:**
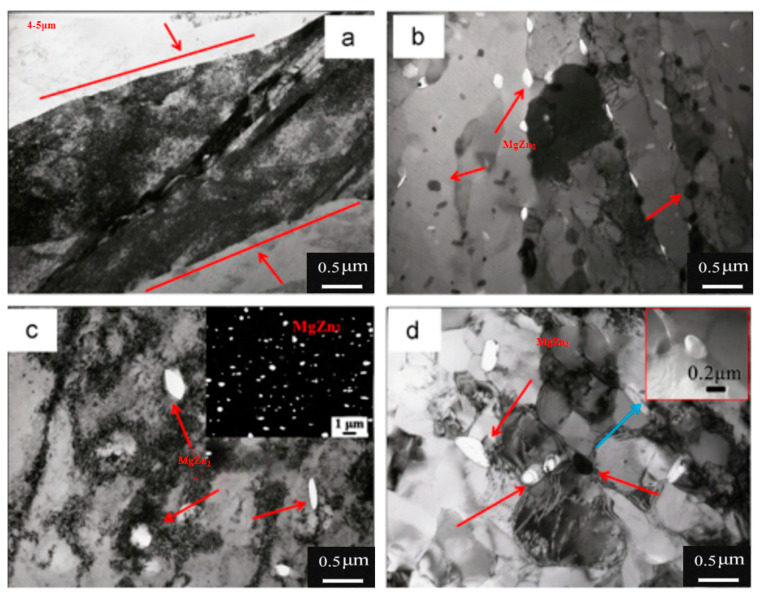
TEM micrographs of 7050 during N-ITMT: (**a**) W + 50% FR; (**b**) W + 50% FR + OA; (**c**) final rolled sheet (the inset in Figure 8c is the distribution of η obtained by SEM); (**d**) bright field micrograph of partially recrystallized 7050 Al (the inset in Figure 8d is dark field micrograph of η). The arrows in all figures correspond to η; white spots in (**b**–**d**) are also η phases etched away during electropolishing [38].

**Figure 9 materials-15-04725-f009:**
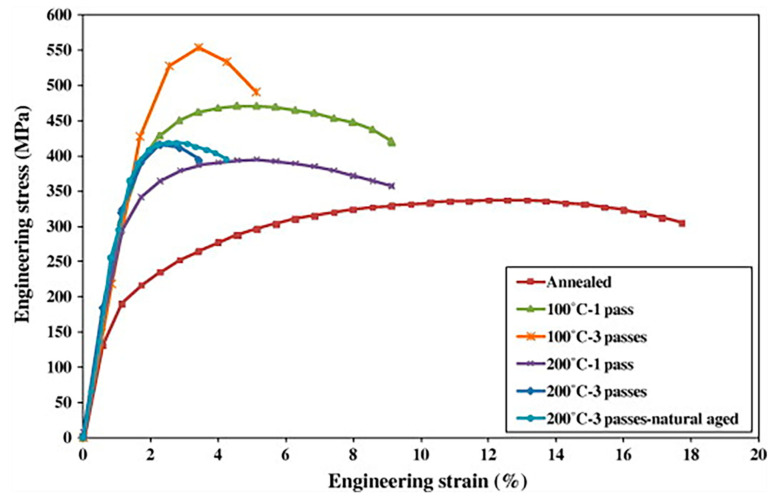
Tensile engineering stress–strain curves for initial and processed materials at room temperature [42].

**Figure 10 materials-15-04725-f010:**
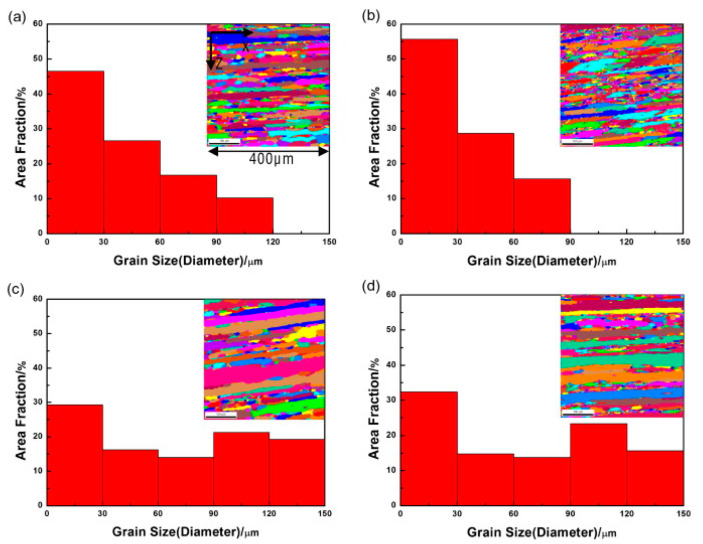
Grain size distribution of A7N01S-T5 alloys with different compositions: (**a**) #1, (**b**) #2, (**c**) #3 and (**d**) #4 [46].

**Figure 11 materials-15-04725-f011:**
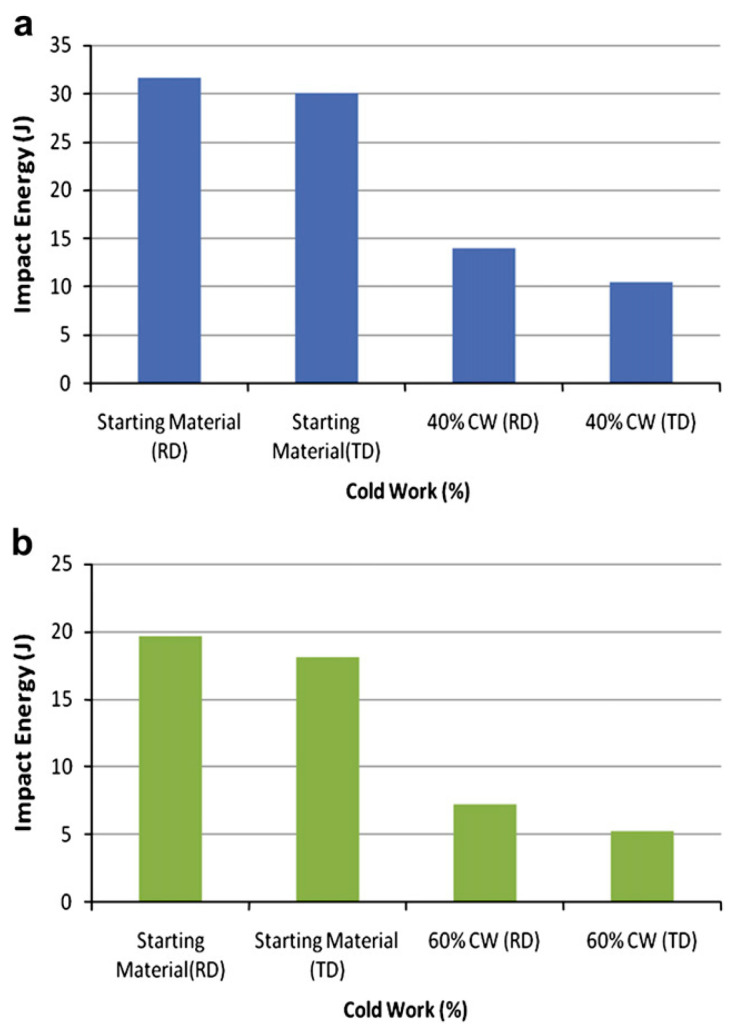
Effects of cold rolling and anisotropy on impact energy of Al alloys: (**a**) 40% cold working (CW) (Plate thickness = 7.5 mm), (**b**) 60% CW (Plate thickness = 5 mm) [48].

**Figure 12 materials-15-04725-f012:**
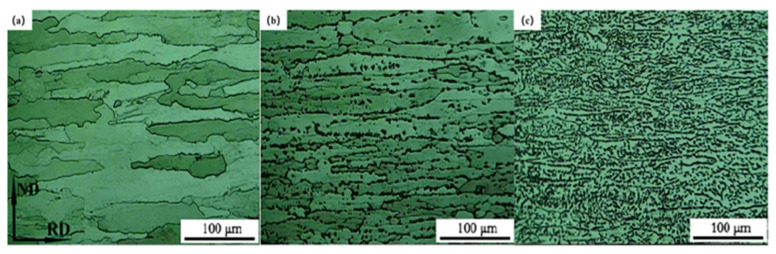
High-strength aluminum alloy samples at (**a**) quenching temperature; (**b**) 405 °C/1 h; (**c**) optical micro-photograph at 340 °C/1 h [68].

**Figure 13 materials-15-04725-f013:**
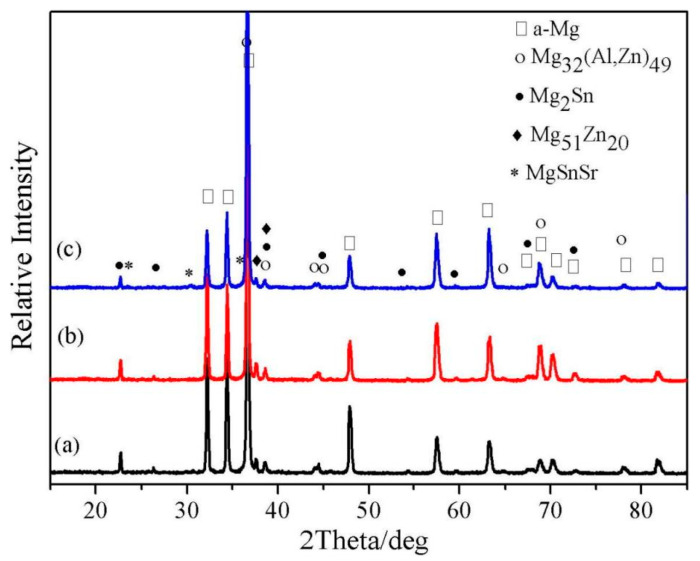
The XRD patterns of as-cast Mg–4.5Zn–4.5Sn–2Al alloys with different Sr additions. (**a**) 0%, (**b**) 0.6% and (**c**) 1.0% [70].

**Figure 14 materials-15-04725-f014:**
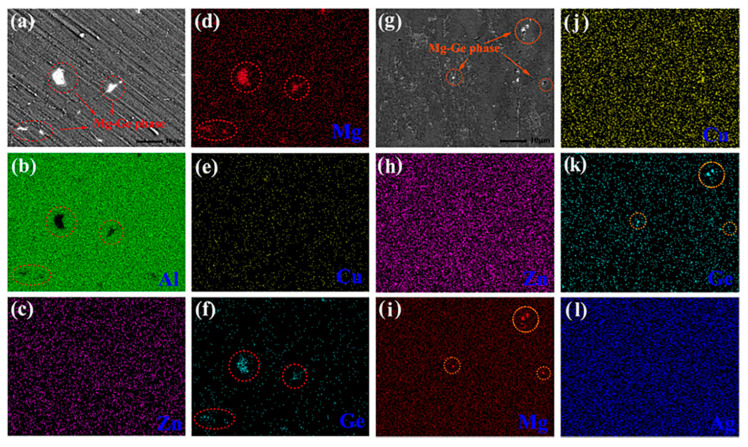
SEM image and element mapping of Ge-added alloy (**a**–**f**) and (Ge+Ag)-added alloy (**g**–**l**) subjected to aging at 120 °C for 25 h after air cooling. The Ge-containing particles are marked by red circles. [71].

**Figure 15 materials-15-04725-f015:**
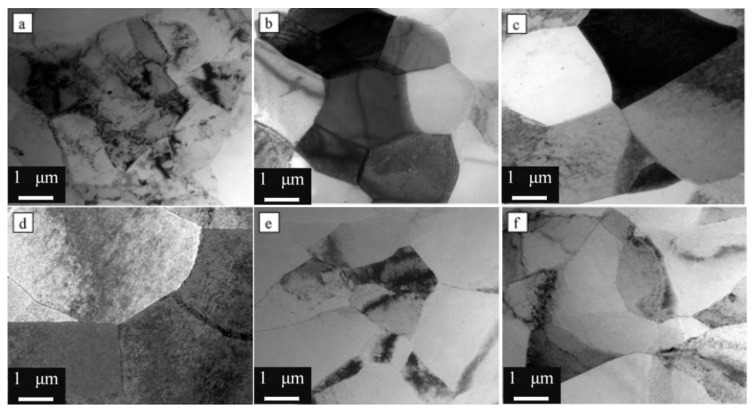
TEM micrographs of sub-grains of 7050 aluminum alloys under different heat treatment conditions: (**a**) SST440, (**b**) SST460, (**c**) SST470, (**d**) SST490, (**e**) EST and (**f**) HTPT [6].

**Figure 16 materials-15-04725-f016:**
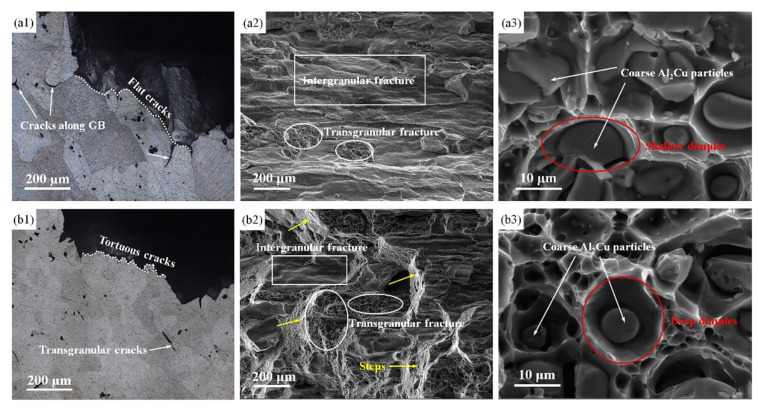
OM and SEM images of axial tensile fracture surfaces of 2219 aluminum alloys: (**a1**–**a3**) HC and (**b1**–**b3**) HC and CD processing [82].

**Figure 17 materials-15-04725-f017:**
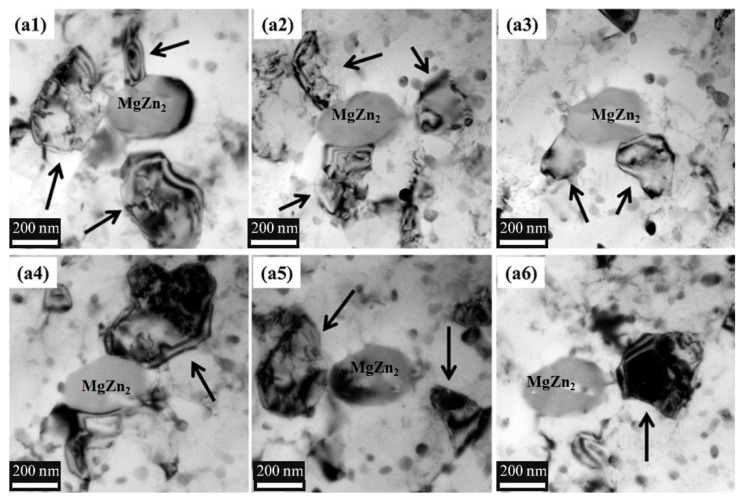
Preferential nucleation sites for recrystallization at large MgZn_2_ grains (black arrows indicating well-developed subgrains; (**a1**–**a6**) show the same BF image with tilted specimen stages) [83].

**Figure 18 materials-15-04725-f018:**
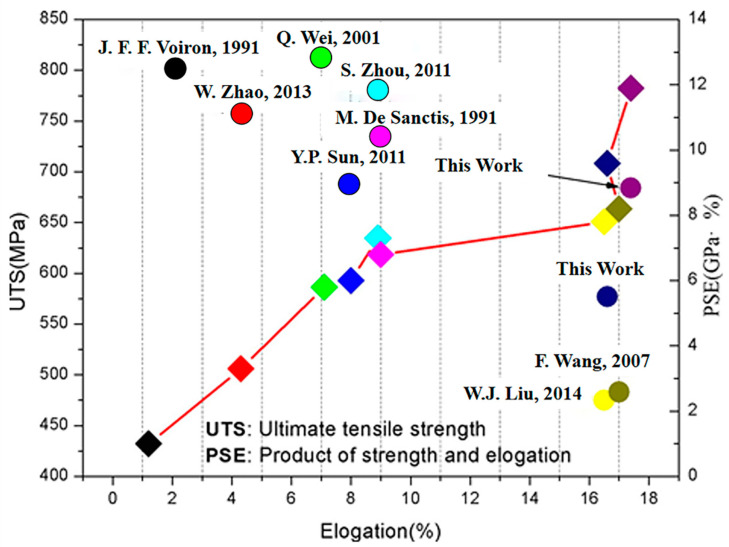
The mechanical properties of different processing methods [92].

**Figure 19 materials-15-04725-f019:**
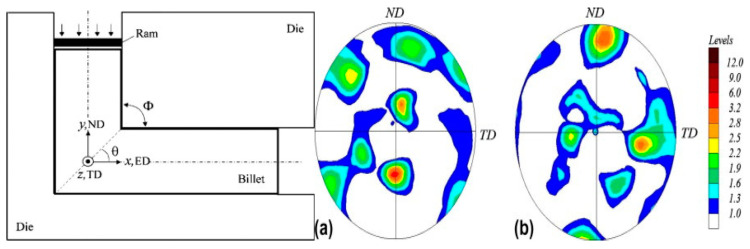
(**a**) ECAP die geometry and coordinate system (1 1 1) and (**b**) (2 0 0) pole figures of Al 7075 alloy subjected to 4 passes of ECAP process by route BC on ED (z) plane [95].

**Figure 20 materials-15-04725-f020:**
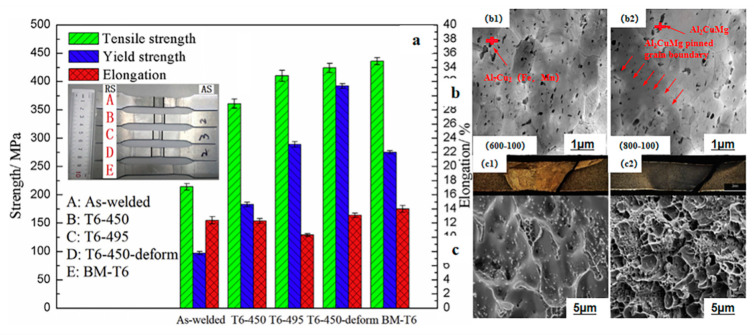
(**a**) Tensile properties of FSW joints under different heat treatment processes (800 rpm, 100 mm/min). (**b**) TEM images of the joints: (**b1**) FSW joints with 600 rpm and 100 mm/min, (**b2**) FSW joints with 800 rpm and 100 mm/min. (**c**) Appearances of fracture surfaces for FSW joints: (**c1**) T6-495 joint, (**c2**) T6-450 deformed joint [101].

**Figure 21 materials-15-04725-f021:**
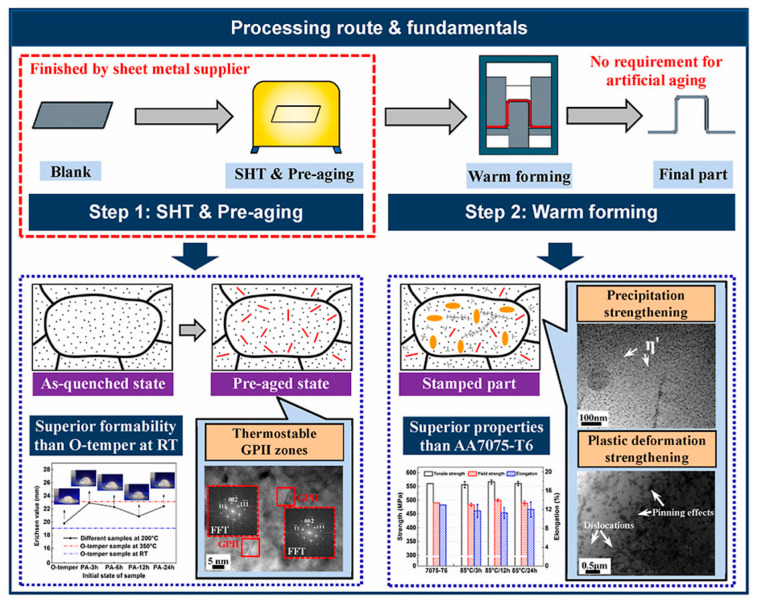
Pre-aged hardening warm forming (PHF) process route and fundamentals [104].

**Figure 22 materials-15-04725-f022:**
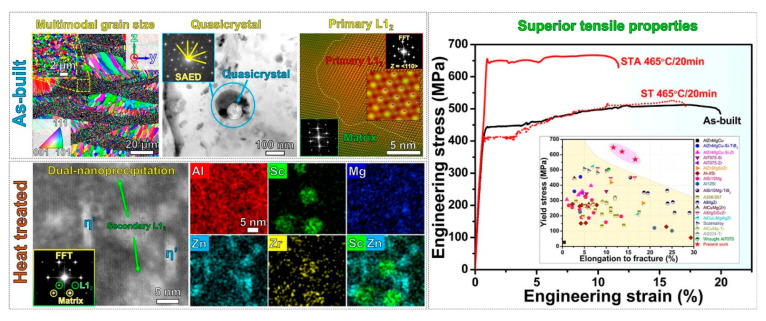
The engineering stress–strain curves of aluminum alloys prepared via SLM under different heat treatment processes, compared with the properties of other types of aluminum alloys. The microstructures of printed and annealed aluminum alloys were analyzed via electron microscope [110].

**Figure 23 materials-15-04725-f023:**
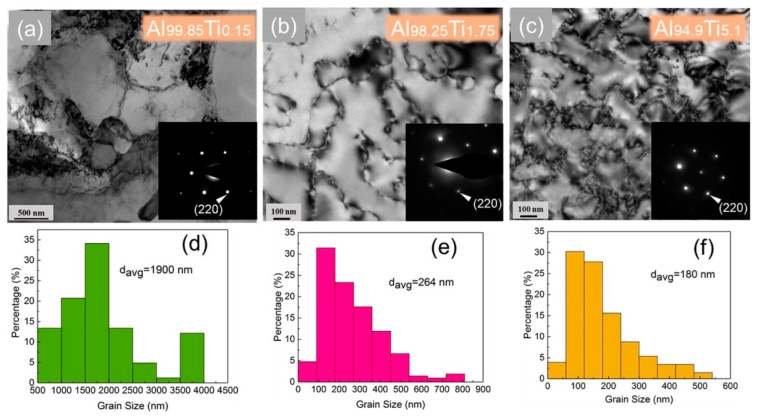
(**a**–**c**) Plan-view TEM micrographs and (insets) selected area diffraction (SAD) patterns of Al–Ti alloy films with various compositions showing the formation of (111) highly textured films. (**d**–**f**) Statistics of grain size distributions show substantial grain refinement, from 1900 to 180 nm with increasing Ti concentrations [112].

**Table 1 materials-15-04725-t001:** Characteristic capacities, key fabrication technologies, and characteristic microstructure of aluminum alloy [15].

Stage	Capacities	Key Technologies and Characteristic Microstructure	Typical Aluminum Alloy
1st generation 1930s~1950s	Static intension	Cr, Mn additions Coherent/semi-coherent precipitates	2024-T4 7075-T6 2618
2nd generation 1950s~1960s	Stress corrosion cracking resistance, damage tolerance	Over-aging Grain-boundary precipitates	7075-T76/T74
3rd generation 1970s~1980s	High strength, corrosion resistance	Purifying, Zr additions Fine constituent particles	7050-T74 2090/2091
4th generation 1990s	High strength, corrosion resistance, more damage tolerance	Further purifying, three-step aging; Discontinuous distribution of grain, narrow precipitation-free zone (PFZ)	7150-T77 7055-T77 2095/2195
5th generation 2000s~now	High strength, low quench sensitivity	Lowering solvus, high-density metastable phases	2099/2199 2050/2060

**Table 2 materials-15-04725-t002:** Elemental composition of tested A7N01S-T5 alloys (wt.%) [46].

Sample No.	Si	Fe	Cu	Factor A		Factor B		Factor C		Al
				Zn	Mg	Mn	Cr	Zr	Ti	
#1	0.11	0.15	0.08	4.34	1.43	0.27	0.13	0.12	0.07	Bal.
#2	0.09	0.15	0.08	4.33	1.47	0.36	0.24	0.16	0.03	Bal.
#3	0.08	0.16	0.08	4.69	1.63	0.22	0.14	0.17	0.03	Bal.
#4	0.09	0.16	0.07	4.54	1.59	0.34	0.24	0.13	0.09	Bal.

## Data Availability

Data sharing is not applicable. No new data were created or analyzed in this study. Data sharing does not apply to this article.

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
