# Peer review of "Recent Progress on Regulating Strategies for the Strengthening and Toughening of High-Strength Aluminum Alloys"

_materials, 2022, doi:10.3390/ma15134725_

Round 1
Reviewer 1 Report
The article on the topic "Recent progress on regulating strategies for strengthening and toughening of high-strength aluminum alloys" well-written review of works and has scientific potential. In my opinion, the work can be published, but after the following corrections:
- Link 24 in the References signed up for two at once (24 and 25). As a result, all further links have been moved one up and do not correspond to the indicated numbering in the text.
- From line 117 missing a space before "[23]”, line 86, 172, 174, 180, 181, 186, 193, 199, 262, 267, 268, 272, 382, 400 (D.(not space)M.(not space)Liu), 476, 630 likewise. From lines 302, 305, 359 (N. (not point) Kamp)), 367, 718 it will be right if “et al.” before literature references. From line 324 incorrect author's name (correct is C. M. Cepeda-Jiménez) and missing a space after "[47]”. From line 553 “N.(not space)M.(not space)Han”. From line 526, 605 and 622 is correct “B. B. Jiang et al. [74]”, “C. Si [92]” and “J. Li [95]”. From line 639 and 718 “Z. L(not point) Hu et al. [100]” and “Z. G(not point) Zhu [110] et al.”.
- Figure 3 increase the font of the text by 2 times. Line 90 in the text indicates the grades of aluminum alloys 2024 and 7075, so duplicate them in Figure 3, since it has a different marking. Also, agree on the marking of alloys in Figure 3 and in Table 1. Duplicate them both in the figure and in the table.
- First point (line 129-130): "During the development of HSAA, there is a mismatch between strength and fracture toughness, and corrosion resistance." Corrosion resistance in the proposal as a prefix to differences in mechanics, however, it should depend on the composition of the phases in the alloy, if the composition is highly heterogeneous, then the barrier properties will be low... Specify how you assume these discrepancies in the key of corrosion resistance... i.e you need to supplement this sentence so that everything is clear...
- Improve the quality of Figure 4, i.e. make it more contrast and increase the font of the text by 1.5 times.
- From lines 172-174 add spacers before abbreviations (SSS, DS, FGS, SPS).
- From line 185 Eq. 1 rename as formula (1) and in Eq. 1 replace the multiplication symbols with standard, i.e reduce them by 3 times, formule 4 (line 376) likewise.
- In the caption to Figure 9, replace dot with space "at.room temperature" and delete point before [42]. In the caption to Figure 10, write the grade of the alloy in capital letters, as in the text.
- In paragraph 3.3, all the numbering of formulas must be aligned to the right edge of the old page, for formula 1 likewise. From lines 439-441 commas are displayed incorrectly.
- It is worth bringing to one design of the unit of measurement, some have spaces after numbers, some do not. I would advise not to put a space before %, but for all the rest, for example: MPa-GPa, ℃, nm-µm, etc. - put. Also, all digits for the phases are in lower case (MgZn2) and al-2.5at%(remove spaces)Fe (lines 749, 756-757).
- Figures 10, 11, 14, 15, 17, 19 have completely illegible lines with units of measurement, it is worth circling them and increasing the font of the inscriptions by 3-5 times. Figures 10, 11, 14, 15, 17, 21, 22 run off the page or are not aligned in the middle of the page.

Author Response
We are truly grateful to your critical comments and thoughtful suggestions. Thank you very much for your kind comments and helpful suggestions. The comments and suggestions are valuable and very helpful for improving the quality of the manuscript. Based on these comments and suggestions, we have made careful modifications on the original manuscript. All changes made to the text are in red color. We hope that the new manuscript will meet your magazine’s standard. Below you will find our point-by-point responses to the Reviewers' comments/questions:Please see the attachment.

Reviewer 2 Report
Overall, the area is worthy of investigation and the authors have reviewed significant relevant literature. The following are the comments for the authors to improve the quality of the manuscript before acceptance.
1. Any academic paper must be sound from both scientific and presentation (academic writing and language) perspectives. The scientific contents of this manuscript are sound; however, there are deficiencies from the presentation perspective. The paper is full of typos, English, and grammar related mistake, so much so, that it is many a times difficult to even understand what authors’ viewpoint is. Some commonly repeating such mistakes are
a. The authors have often used very long sentences and on top of that this is combined with wrong/unnecessary used of colons and semicolons which further increases the sentences length. Make sure no sentence is longer than 2.5 lines or in rare cases max 3 lines. Below I am mentioning a few such examples
i. In abstract “This paper first analyzes the problems….systematically emphasized” 9 lines = one sentence!!
j. Line 48 – 52 is a single sentence., Line 443 to 448 is a single sentence.
k. Please carefully check all the manuscript for such long sentences and rephrase them into smaller easy to read and understandable sentences.
b. Often the authors do not add a space between the preceding word the bracket. E.g. line 117 “…sensitivity[23]..”, line 324 “…Nez [47]and..”, Figure 18 caption, “(b)Porosity”, Figure 20 caption, “(a)Tensile”
c. HSAA is a plural but sometimes the authors deal it as singular.
d. I highly recommend that paper must be proof read by some professional English editing services.
2. The intro section is too short and finish too abruptly after discussing only a few studies and does not even mentioned what is going to be discussed in the following.
3. The claim of the authors that they have discussed the numerical simulations and their experimental verifications is not very true. Infact, the authors have only discussed some formulas for predicting the strength of the HSAA. This part of the paper is very weak, either remove is or expand it properly. Better to remove it as authors did added any conclusion for it either in section 6.
4. Most of the figures in the paper are of very poor quality and text is barely readable in them. For example, Figure 4 must be improved. A simple way to do this is to add the same text by yourself in the figures.
5. Some figure and labels of sub-figures but there is no figure caption provided for the subfigures, e.g., Fig. 8, 15, 17…check other please.
6. Conclusions are not very convincing and must be improved and expanded a little more. Line 772 to 776 is not a conclusion. At the moment the conclusions are more general, please add more specific conclusions.
Author Response
We are truly grateful to your critical comments and thoughtful suggestions. Thank you very much for your kind comments and helpful suggestions. The comments and suggestions are valuable and very helpful for improving the quality of the manuscript. Based on these comments and suggestions, we have made careful modifications on the original manuscript. All changes made to the text are in red color. We hope that the new manuscript will meet your magazine’s standard. You will find our point-by-point responses to the Reviewers' comments/questions.Please see the attachment.

Round 2
Reviewer 2 Report
Thank you for your revisions. By the way, you mentioned at the end of the cover letter that "We wish this paper can be published in “the International Journal of Advanced Manufacturing Technology”." I think this is not the case!
Author Response
We are truly grateful to your critical comments and thoughtful suggestions. Thank you very much for your kind comments and helpful suggestions. The comments and suggestions are valuable and very helpful for improving the quality of the manuscript. Based on these comments and suggestions, we have made careful modifications on the original manuscript. All changes made to the text are in red color. We hope that the new manuscript will meet your magazine’s standard.
